# Shikonin Reduces Growth of Docetaxel-Resistant Prostate Cancer Cells Mainly through Necroptosis

**DOI:** 10.3390/cancers13040882

**Published:** 2021-02-20

**Authors:** Sascha D. Markowitsch, Kira M. Juetter, Patricia Schupp, Kristine Hauschulte, Olesya Vakhrusheva, Kimberly Sue Slade, Anita Thomas, Igor Tsaur, Jindrich Cinatl, Martin Michaelis, Thomas Efferth, Axel Haferkamp, Eva Juengel

**Affiliations:** 1Department of Urology and Pediatric Urology, University Medical Center Mainz, Langenbeckstraße 1, 55131 Mainz, Germany; sascha.markowitsch@unimedizin-mainz.de (S.D.M.); kjuetter@students.uni-mainz.de (K.M.J.); pschupp@students.uni-mainz.de (P.S.); kristine.hauschulte@gmx.de (K.H.); olesya.vakhrusheva@unimedizin-mainz.de (O.V.); kimberlysue.slade@unimedizin-mainz.de (K.S.S.); anita.thomas@unimedizin-mainz.de (A.T.); igor.tsaur@unimedizin-mainz.de (I.T.); axel.haferkamp@unimedizin-mainz.de (A.H.); 2Institute of Medical Virology, Goethe-University, 60596 Frankfurt, Germany; cinatl@em.uni-frankfurt.de; 3Industrial Biotechnology Centre and School of Biosciences, University of Kent, Canterbury CT2 7NJ, UK; M.Michaelis@kent.ac.uk; 4Institute of Pharmaceutical and Biomedical Sciences, Johannes Gutenberg University Mainz, Staudingerweg 5, 55128 Mainz, Germany; efferth@uni-mainz.de

**Keywords:** prostate cancer (PCa), docetaxel (DX) resistance, shikonin (SHI), Traditional Chinese Medicine (TCM), growth inhibition, apoptosis, necroptosis

## Abstract

**Simple Summary:**

Prostate carcinoma (PCa) is the most common tumor in men with an increasing age-associated risk. Several therapy strategies, one of which is docetaxel (DX) chemotherapy, have been established. However, due to the development of therapy resistance, in which chemotherapy no longer effectively combats the cancer, advanced, metastasized PCa with a poor prognosis may become manifested and therapy inevitably fails. Thus, new treatment options are urgently needed. Shikonin (SHI), from Traditional Chinese Medicine, has revealed promising antitumor activity in several tumor entities. In the current study, the impact of SHI on four therapy-sensitive and four respective DX-resistant PCa cell lines was determined. SHI induced growth inhibition mainly by necroptosis, a type of cell death, in all the tested therapy-sensitive, but more importantly, DX-resistant PCa cell lines. Corresponding molecular alterations contributing to growth inhibition after SHI exposure were found. SHI could, therefore, be a promising additive in treating advanced PCa.

**Abstract:**

The prognosis for advanced prostate carcinoma (PCa) remains poor due to development of therapy resistance, and new treatment options are needed. Shikonin (SHI) from Traditional Chinese Medicine has induced antitumor effects in diverse tumor entities, but data related to PCa are scarce. Therefore, the parental (=sensitive) and docetaxel (DX)-resistant PCa cell lines, PC3, DU145, LNCaP, and 22Rv1 were exposed to SHI [0.1–1.5 μM], and tumor cell growth, proliferation, cell cycling, cell death (apoptosis, necrosis, and necroptosis), and metabolic activity were evaluated. Correspondingly, the expression of regulating proteins was assessed. Exposure to SHI time- and dose-dependently inhibited tumor cell growth and proliferation in parental and DX-resistant PCa cells, accompanied by cell cycle arrest in the G2/M or S phase and modulation of cell cycle regulating proteins. SHI induced apoptosis and more dominantly necroptosis in both parental and DX-resistant PCa cells. This was shown by enhanced pRIP1 and pRIP3 expression and returned growth if applying the necroptosis inhibitor necrostatin-1. No SHI-induced alteration in metabolic activity of the PCa cells was detected. The significant antitumor effects induced by SHI to parental and DX-resistant PCa cells make the addition of SHI to standard therapy a promising treatment strategy for patients with advanced PCa.

## 1. Introduction

Prostate carcinoma (PCa) is the most common cancer and the second leading cause of cancer mortality in men, with increasing risk associated with age. Early detection programs in middle age and new therapeutic strategies are attempts to extend the life expectancy of PCa patients [1]. Despite the advantages of currently approved therapeutic strategies, advanced PCa remains an aggressive disease with poor prognosis. For high-risk, non-metastatic, and metastatic castration-resistant PCa, the efficacy of the chemotherapeutic agent docetaxel (DX) was established [2] and approved as a “first-line” therapy after androgen deprivation therapy (ADT). In the following years, DX was further approved as a first-line therapy in combination with ADT [3]. A good response in high-risk non-metastatic PCa patients to DX applied together with ADT and radiation has been reported [4]. However, therapy resistance limits the effect of DX to a few months and failure inevitably occurs [5]. The efficacy in second-line treatment decreases considerably with therapy resistance [6].

Failure of conventional therapy prompts cancer patients to turn to traditional and alternative medicine [7,8]. Nearly half of cancer patients in Europe use complementary and alternative therapies [9,10,11], hoping to increase effectiveness or reduce the side effects of conventional therapy [12,13]. The turn to complementary medicine is critical, as reliable studies, and thus proven efficacy of natural substances, are often not available, leading to uncoordinated self-treatment. Lack of studies also increases the risk of unidentified contraindications and adverse side effects of the natural compounds combined with conventional therapy [14]. However, some studies have been carried out indicating antitumor effects of natural compounds, notably if applied together with an established therapy or by counteracting therapy resistance [15,16,17,18,19,20,21].

One natural compound that has revealed promising antitumor activity is shikonin (SHI). SHI is a herbal pigment traditionally used for the natural coloration of textiles and food. It is isolated from the dried roots of *Lithospermum erythrorhizon*. Several studies show that SHI, in addition to its function as a color pigment, has antimicrobial [22], antiinflammatory [23], and antitumor [24,25] activity.

The antitumor activity has been demonstrated in different tumor entities. In gastric cancer, cells exposed to SHI underwent cell cycle arrest in the G2/M phase [26]. SHI has induced apoptosis in gastric cancer [27] and pancreatic cancer [28] in vitro. Other types of regulated cell death have also been detected [29]. One of these is necroptosis and exposure to SHI in glioma [30], breast [31], and pancreatic cancer cells [28] has resulted in this type of regulated cell death. Necroptosis is a caspase-independent, programmed form of necrosis [32]. Receptor-interacting serine/threonine-protein kinase 1 and 3 (RIP1/RIP3) are relevant key players in this signaling pathway. Their activation is associated with the inactivation of caspase 8, followed by mixed lineage kinase domain-like protein (MLKL) complex formation and degeneration of cell membranes and organelles [33]. In bladder cancer [34] and chronic myeloid leukemia cells [35], SHI led to a decrease in chemotherapy resistance through induction of necroptosis.

In addition, SHI exerted a direct influence on cell metabolism, specifically inhibiting pyruvate kinase isozymes M2 (PKM2) and directly or indirectly resulting in reduced growth of bladder cancer [34] and lung cancer in vitro [36] and in vivo [37]. SHI inhibited the mitochondrial activity of cancer cells [38]. However, data for SHI in PCa are sparse and not available for DX-resistant PCa cells. Thus, the current study was designed to evaluate the impact of SHI on the growth behavior of therapy-sensitive (parental) and DX-resistant PCa cells.

## 2. Results

### 2.1. Shikonin Inhibited Cell Growth of Parental and DX-Resistant PCa Cells

To determine the SHI concentration necessary to influence cell growth in the four parental and DX-resistant PCa cell lines (PC3, DU145, LNCaP, and 22Rv1), the cells were exposed to SHI ranging from 0.1 to 1.5 µM. A dose- and time-dependent growth inhibition was apparent in all four cell lines, compared to their respective unexposed controls (Figure 1). A significant growth reduction of parental PC3 and DU145 cells with an IC50 of 0.37 µM SHI after 72 h was determined (Figure 1a,c). The growth of DX-resistant PC3 and DU145 cells was inhibited with an IC50 of 0.54 µM and 0.55 µM SHI after 72 h treatment (Figure 1b,d), indicating that growth inhibition of the parental and DX-resistant PC3 and DU145 cells is similar, but slightly stronger in parental cells. However, for both parental and DX-resistant PC3 cells significant growth inhibition was first reached with 0.5 µM SHI (Figure 1a,b). For parental and DX-resistant DU145 cells, considerable inhibition was achieved with 0.75 µM SHI after 72 h incubation (Figure 1b,c). For LNCaP cells, it was the other way around. Here, with an IC50 of 0.59 µM (Figure 1d), parental LNCaP showed a lower response to SHI, comparable to DX-resistant PC3 and DU145 cells. The DX-resistant LNCaP counterpart displayed a higher sensitivity with an IC50 of 0.32 µM (Figure 1e). 22Rv1 revealed the lowest sensitivity to SHI with an IC50 of 1.05 µM in the parental and an IC50 of 1.12 µM in the DX-resistant cells after 72 h (Figure 1g,h). Statistically significant growth inhibition in 22Rv1 was reached only with the highest dose of 1.5 µM SHI (Figure 1h). Thus, for further investigation into the mechanisms responsible for the growth inhibitory activity of SHI, 0.5 µM to 1.0 µM SHI was used for PC3, DU145, and LNCaP cells and 0.5–1.5 µM SHI for the 22Rv1 cells.

### 2.2. Shikonin Impaired PCa Cell Proliferation

Application of SHI for 24 and 48 h resulted in a dose-dependent inhibition of proliferation in all four investigated PCa cell lines (Figure 2). Analogous to growth, parental PC3 and DU145 cells showed a higher sensitivity to SHI, compared to their DX-resistant counterparts (Figure 2a–d). Most of the PCa cells already revealed strong or most potent effects after 24 h SHI treatment. However, parental DU145 cells responded better to SHI after 48 h exposure. In the DX-resistant subcells, first significant antiproliferative events were only apparent after 48 h SHI application (Figure 2c,d). In contrast, treatment of both parental and DX-resistant LNCaP cells with 0.5 µM SHI resulted in a significant inhibition of proliferation at all measured time points (Figure 2e,f). Proliferation was further suppressed with a higher SHI concentration. Similar to the growth experiments, 22Rv1 exhibited the lowest sensitivity to SHI, and the highest concentrations of 0.75 µM and 1.0 µM SHI were necessary to significantly reduce the proliferation of parental and DX-resistant 22Rv1 cells (Figure 2g,h).

### 2.3. Shikonin Induced Cell Cycle Arrest and Alterations in the Expression and Activity of Cell Cycle Regulating Proteins

Diminished tumor cell growth and proliferation after SHI treatment were partially due to impaired cell cycle progression (Figure 3). Exposure to 0.5 µM SHI provoked a significant increase of cells in the G2/M phase in parental PC3 and DX-resistant DU145 cells. This was associated with a decrease of G0/G1 phase cells in parental PC3 and a reduction of S phase cells in DX-resistant DU145 (Figure 3a,d). DX-resistant 22Rv1 cells showed a significant elevation of the S phase after exposure to SHI, independent of the SHI concentration. This was accompanied by tendency by a decrease of cells in the G0/G1 and G2/M phases.

DX-resistant PC3, parental DU145, parental 22Rv1 cells, as well as parental and DX-resistant LNCaP cells showed no significant changes in cell cycle progression after SHI exposure (Figure 3b–g). This indicates that other mechanisms are responsible for the observed inhibition of tumor cell growth and proliferation in these PCa cells.

As exposure to SHI revealed the strongest effects on the cell cycle progression in PC3 and DU145 cells, the expression and activity of cell cycle regulating proteins in these cell lines were evaluated. Indeed, modulation of the cell cycle phases was accompanied by significant alteration in the cell cycle regulating proteins (Figure 4, Figure 5 and Figure 6, and Appendix A). Exposure to SHI resulted in a significant accumulation of p21 in parental but not in DX-resistant PC3 cells (Figure 4 and Figure 5a, and Appendix A). In addition, there was a significant decrease of p27 in parental cells, whereas DX-resistant cells showed by tendency an increase (Figure 4 and Figure 5b, and Appendix A). Cyclin A, B and Cyclin-dependent kinase (CDK) 1, essential for G2/M phase [39], were significantly reduced by SHI in both parental and DX-resistant PC3 cells (Figure 4 and Figure 5c,d,g, and Appendix A). Furthermore, together with Cyclin A, the expression of CDK2, both responsible for regulating S phase progression [40], was significantly reduced by exposure to SHI (Figure 4 and Figure 5f,g, and Appendix A). The active, phosphorylated form of CDK2 was also significantly decreased after exposure to SHI (Figure 4 and Figure 5h, and Appendix A). Cyclin D1, involved in G0/G1 phase progression, was not affected (Figure 4 and Figure 5e, and Appendix A). In good accordance with the cell cycle arrest of DX-resistant DU145 cells in the G2/M phase, alterations in the expression and activity of cell cycle-regulating proteins were mainly seen in DX-resistant cells (Figure 4 and Figure 6, and Appendix A). This included the proteins Cyclin B, CDK1, CDK2, and pCDK2 (Figure 4 and Figure 6d,f–h, and Appendix A). However, CDK1 and 2 protein levels were also significantly reduced in parental DU145 cells (Figure 4 and Figure 6f,g, and Appendix A).

### 2.4. Shikonin Induced Cell Death

Exposure to SHI dose-dependently resulted in an accumulation of apoptotic events in parental and DX-resistant PC3, and DU145 cells (Figure 7a,b). 22Rv1 cells displayed less pronounced cell death (Figure 7d). No apoptosis was detectable in parental LNCaP cells, and only applying a higher concentration of 0.7 µM SHI or more contributed to apoptosis in the DX-resistant LNCaP cells (Figure 7c). Consequently, other antitumor effects of SHI must be responsible for the observed inhibition of growth and proliferation. Necrotic events were not apparent in the PCa cells.

Due to the low sensitivity of the 22Rv1 cell lines to SHI, further investigation was directed towards parental and DX-resistant PC3, DU145, and LNCaP cells. As a caspase-dependent cell death could account for the growth inhibition induced by SHI, cells were exposed to zVAD, a multi-caspase inhibitor. However, in combination with SHI, zVAD did not influence the growth of parental and DX-resistant PC3, DU145, and LNCaP cells (Figure 8a–f), indicating a caspase-independent cell death induction. Parental DU145 cells treated with 12.5 nM DX were used as a positive control, as DU145 has been shown to respond with a caspase-dependent apoptosis initiation after DX application [41]. In fact, combined treatment with DX and zVAD led to a significant recovery in tumor cell growth (Figure 8g). No changes in the protein expression of PARP or caspase 3 were apparent after SHI exposure, neither in parental nor in DX-resistant DU145 cells (Figure 8h,i and Appendix A), further corroborating the hypothesis of a caspase-independent cell death. Notably, the expression of caspase 8 significantly decreased after exposure to SHI (Figure 8j and Appendix A), indicating a caspase-independent cell death induction, such as necroptosis.

### 2.5. Shikonin Induced Necroptotic Effects

Necroptosis is a caspase-independent cell death and necrostatin-1 inhibits the activity of RIP1 and blocks the necroptosis pathway. As SHI induced necroptosis in various tumors [28,30,32,34], necrostatin-1 was applied to determine whether SHI also has an impact on PCa tumor cell growth. SHI application significantly reduced growth in all cell lines (Figure 9a–f). Combined administration of 0.5–1.0 µM SHI and 80 µM necrostatin-1 resulted in a reversal of SHI’s antigrowth effect in all parental and DX-resistant PCa cell lines, leading to cell growth comparable to the untreated controls (Figure 9a–f).

Representative for the tested PCa cell lines, PC3 and DU145 cells showed an increase in pRIP1 and/or pRIP3 activation after exposure to SHI (Figure 10b,d,g,i and Appendix A). In parental PC3, pRIP1 and pRIP3 were significantly activated by SHI, whereas additional application of necrostatin-1 reversed this activation (Figure 10b and Appendix A). DX-resistant PC3 cells revealed no effect on pRIP1 after exposure to SHI but displayed by tendency an elevation of pRIP3, compared to the SHI-untreated controls (Figure 10d and Appendix A). Again, combined treatment with SHI and necrostatin-1 counteracted this activation and led to a significant decrease of pRIP1 and pRIP3, compared to the SHI-treated cells. In the DU145 cells both parental and stronger DX-resistant DU145 cells showed a significant upregulation of pRIP1 by SHI (Figure 10g and Appendix A). Addition of necrostatin-1 to SHI in parental and DX-resistant DU145 cells significantly abolished RIP1 phosphorylation. pRIP3 was also significantly amplified after SHI application in parental DU145 cells (Figure 10i and Appendix A). As before, phosphorylation was abrogated by combining SHI with necrostatin-1. In contrast, the expression of total RIP1, RIP3, and MLKL was not significantly affected by SHI (Figure 10a,c,e,f,h,j and Appendix A), and pMLKL was not detectable in the PC3 and DU145 cells. Combined application to SHI and necrostatin-1 significantly reduced the total amount of RIP3 in parental PC3 (Figure 10c and Appendix A) and of RIP1 in DX-resistant DU145 (Figure 10f and Appendix A).

In addition, administration of 0.5 µM SHI resulted in a significant decrease of the GSH-content in parental and DX-resistant DU145 cells (Figure 11), indicating ROS generation.

### 2.6. Shikonin Showed No Effects on Metabolism

SHI has been shown to directly influence mitochondrial activity [38] and serve as a specific pyruvate kinase M2 inhibitor [24]. Inhibiting this enzyme with SHI could therefore directly influence tumor cell metabolism. Basal oxygen consumption rate and extracellular acidification rate were comparable in parental and DX-resistant PCa cells. However, treatment with SHI resulted only in temporarily elevated mitochondrial respiration and decreased aerobic glycolysis in the DU145 cells, indicating transient enhanced oxidative phosphorylation in response to a stress stimulus (Appendix A).

## 3. Discussion

Prostate carcinoma is the most common malignant tumor in men. Currently, there is no curative therapy for advanced prostate carcinoma, and palliative treatment is most often the only open option. Conventional therapeutic approaches are intended to prolong progression-free survival but are limited in their effect and result in resistance, so that new treatment strategies are crucial. Addition of SHI is a possible treatment strategy, as in the current study it inhibited growth and reduced proliferation of four parental and DX-resistant PCa cell lines. In good accordance with this, SHI treatment in lung [42], gallbladder [43], esophagus [44], and breast cancer [45] has resulted in reduced growth in vitro. SHI has also shown in vivo growth inhibition of nasopharyngeal cancer [46] and melanoma [37]. Diminished growth with SHI has also been shown in therapy-sensitive PC3 and LNCaP cells [47] through inhibition of the AKT/mTOR signaling pathway. Exposure to SHI has been shown to restrict the growth of LNCaP, and 22Rv1 cells by affecting the androgen receptor [48].

Growth and proliferation inhibition after treatment with SHI was associated with a cell cycle arrest of parental PC3 and DX-resistant DU145 cells in the G2/M phase, and an S-phase arrest in DX-resistant 22Rv1. Furthermore, in pancreatic, lung [25], breast [31], gastric cancer cells [26], and melanoma [49] the administration of SHI resulted in cell cycle arrest in the G2/M phase. In a study on therapy-sensitive PC3 and DU145 cells, SHI induced a shift to the G2/M phase [47]. However, data of cell cycle regulating proteins were missing.

In the current study, the SHI-initiated G2/M phase arrest in PC3 and DU145 cells was evident at the protein level. In parental and DX-resistant PC3 cells, SHI induced a significant decrease in the cell cycle regulating proteins Cyclin A, Cyclin B, CDK1, and CDK2, which are responsible for G2/M phase progression, whereas an increase of p21 and a decrease of p27 was only apparent in parental PC3 cells. Consistent with the cell cycle data, DX-resistant DU145 cells showed a stronger downregulation of Cyclin B, CDK1, and CDK2 by SHI, compared to the parental cells. In line with the current data on PCa cells, treatment of gastric cancer cells with SHI resulted in a G2/M phase arrest, associated with a reduction of cell cycle activating proteins and an increase in the cell cycle inhibitor p21 [26]. Reduced Cyclin B and increased p21, concomitant to a G2/M arrest after SHI application, have also been detected in melanoma cells [49]. Downregulation of p27 triggered cell cycle arrest in endothelial cells [50] and pancreas cancer cells [51], in connection with a tumor suppressor activity. As Cyclin B in complex with CDK1, mediates the transition from the G2 to M phase [52] the downregulation of Cyclin B by SHI may prevent the transition from the G2 to M phase. This may produce the cell cycle arrest in the parental and DX-resistant PCa cells observed here.

Aside from the cell cycle arrest in the G2/M phase, growth inhibition was accompanied by a significant apoptosis increase in all the PCa cell lines, except for LNCaP, where only a higher dose of SHI induced significant apoptotic effects in the DX-resistant cells. Necrotic events after SHI exposure were not detectable in the parental and DX-resistant PCa cells; thus, necrosis induction could not be responsible for SHI’s growth inhibitory effect. In good accordance with the current apoptosis data, SHI induced apoptosis in various tumor entities. In esophageal cancer cells, apoptosis induction by SHI was a result of the specific inhibition of PKM2, leading to a loss of energy generation [53]. In contrast, in lung cancer cell lines the apoptosis induction by SHI has been attributed to the FOXO3a/EGR1/SIRT1 pathway [42]. In leukemia cells, SHI led to apoptosis, postulated to be associated with the inhibition of c-Myc [54]. Apoptosis could also be induced in gastric cancer cell lines via the mitochondrial caspase-dependent pathway, as shown by the multi-caspase inhibitor zVAD, which inhibits the antigrowth effect of SHI [27].

However, in the current study zVAD did not abolish the inhibitory effect of SHI on growth, indicating a caspase-independent induction of cell death. In fact, zVAD even significantly decreased the inhibitory effect of DX on DU145 cell growth. It is known that DX induces caspase-dependent apoptosis in DU145 cells [41]. However, in good accordance with the current SHI data regarding PCa, zVAD did not block the growth inhibitory effect of SHI in lung cancer cells [29]. Similar results have been obtained with osteosarcoma [55], and gastric cancer cells [28]. Here too, growth inhibition by SHI could not be reverted by zVAD, indicating an absence of caspase-dependent apoptosis. Indeed, exposure to SHI alone in bladder carcinoma [34] and osteosarcoma cells [55] had no impact on the expression of caspase 3. Accordingly, in the PCa cells the expression of caspase 3 was not altered by the application of SHI, further corroborating the hypothesis of a caspase-independent apoptosis induction.

Notably, caspase 8 was significantly reduced in the PCa cells after exposure to SHI, suggesting a necroptosis induction, an apoptosis-related programmed cell death [32]. Necroptosis is a regulated cell death, but in contrast to “classic” apoptosis, which is known to be caspase 3-dependent, necroptosis is characterized by diminished caspase 8 expression and activity. Activated caspase 8 would lead to apoptosis induction. Inactivation and downregulation of caspase 8 plays a pivotal role in necroptosis, facilitating the formation of a necrosome complex, consisting of RIP1, RIP3, and MLKL. This complex leads to membrane permeabilization and finally to cell death.

Indeed, adding the necroptosis inhibitor necrostatin-1 resulted in a tumor cell growth equivalent to the untreated controls in all parental and DX-resistant PCa cells, including LNCaP. Accordingly, in gastric cancer cell lines, necrostatin-1 blocked the antitumor effect of SHI [56]. In lung cancer cells, administration of SHI and necrostatin-1 resulted in a significant reversion of SHI’s inhibitory effect [29]. Notably, the combined treatment with SHI, necrostatin-1, and DX in the DX-resistant PC3 and DU145 cells displayed even increased growth beyond the growth of the SHI and necrostatin-1 untreated controls. Taxanes, such as DX, induced necroptosis in lung [57] and breast cancer [58,59]. Thus, at least in the DX-resistant PC3 and DU145 cells, necroptotic processes seem to be enhanced after the combined exposure to SHI and DX, which might indicate re-sensitivation of the DX-resistant PCa cells. Pancreatic cancer treatment with SHI and gemcitabine [28] as well as combined application of SHI and erlotinib to glioblastoma induced synergistic effects [60]. Furthermore, chronic administration of SHI with cisplatin or paclitaxel to different cancer cell lines prevented resistance induction [61]. These studies with SHI application to other tumor entities seem to further corroborate our hypothesis that SHI reactivates the necroptotic activity of DX.

RIP1 and RIP3 are critical proteins involved in necroptosis induction [32]. Consistent with this, S166 phosphorylation of RIP1, indicating induction of necroptotic signaling, was evident in parental PC3 and DU145 cells, as well as in DX-resistant DU145 and by tendency in PC3 cells after SHI treatment. Furthermore, in gastric cancer cell lines [28], glioma [62], and osteosarcoma cells [55] administration of SHI resulted in a significant increase of RIP1 and RIP3. In the parental PCa cells, pRIP1 facilitated phosphorylation of RIP3, a downstream effector of the necrosome complex. Elevated phosphorylation of RIP1 and RIP3 was reversed when necrostatin-1 was added. pMLKL is another component of the necrosome complex downstream of RIP3 [63]. MLKL is recruited and phosphorylated by pRIP3, the next step in initiation of necroptosis. In the PCa cells, no pMLKL was detectable after exposure to SHI, as a downstream target probably occurring after the chosen 12 h incubation. The application period might also explain why the parental PCa cells showed stronger effects in the phosphorylation of RIP3 than the DX-resistant cells, although necroptotic effects were more pronounced - but at a later time point. However, after 12 h SHI treatment, pRIP1 and pRIP3 were upregulated in the parental and by tendency in the DX-resistant PCa cells, further confirming the postulated functional role of SHI in necroptosis initialization.

The GSH-content was also significantly diminished after SHI application in the parental and DX-resistant PCa cells, indicating ROS generation. Necroptosis induction has also been shown to be accompanied by increased ROS levels in nasopharyngeal carcinoma cells [46]. SHI-induced GSH depletion and intracellular ROS increase in tumor cells has been demonstrated to be RIP1- and RIP3-mediated [62,64] as well, further confirming that SHI induces necroptosis, as observed in the current investigation.

Therefore, the measured “apoptotic” effects mainly seem to be due to necroptosis. However, the SHI treatment of LNCaP cells revealed only marginal apoptosis, indicating another mechanism. In contrast to the other tested PCa cells, LNCaP cells are androgen receptor (AR)-positive and androgen-sensitive [65,66]. As SHI inhibited AR [48], which prevents cell death processes through the tumor necrosis factor-α (TNF-α) [67], this inhibition might play a crucial role in the necroptosis induction in LNCaP. Notably, TNF-α is involved in necroptotic processes [68,69]. However, the role of TNF-α in LNCaP cells requires further investigation.

SHI has also been described to directly or indirectly influence the metabolism of cancer cells [38,70]. In the current investigation, SHI only induced a significant short-term increase in OCR in the DU145 cells, partially accompanied by decreased glycolysis. This short-lasting metabolic shift towards mitochondrial respiration might indicate a temporary avoidance of apoptosis induction, as has previously been hypothesized [71]. Necroptosis induction by SHI has been postulated to overcome apoptosis resistance [34,72]. Indeed, SHI-induced necroptosis prevented tumor escape, resulting in significant growth inhibition of the PCa cells.

The androgen-insensitive PC3 and DU145 cells showed the highest sensitivity to SHI, whereas LNCaP exhibited the lowest sensitivity. However, in all four parental and DX-resistant PCa cell lines investigated here, SHI induced significant growth inhibition and necroptosis, accompanied by corresponding alterations in cell cycle and cell death regulating proteins. Further investigations in vitro and in vivo are necessary to verify this in vitro data.

## 4. Materials and Methods

### 4.1. Cell Cultures

Prostate cancer cell lines PC3, DU145, LNCaP, and 22Rv1 were obtained from the German Collection of Microorganisms and Cell Cultures (DSMZ). The DX-resistant sublines were derived from the Resistant Cancer Cell Line (RCCL) collection (https://research.kent.ac.uk/industrial-biotechnology-centre/the-resistant-cancer-cell-line-rccl-collection/) [73]. LNCaP cells were grown and subcultured in Iscove Basal medium (Biochrom GmbH, Berlin, Germany), and PC3, DU145, and 22Rv1 cells were grown in RPMI-1640 medium (Gibco, Thermo Fisher Scientific, Darmstadt, Germany). Media were supplemented with 10% fetal calf serum (FCS) (Gibco, Thermo Fisher Scientific, Darmstadt, Germany), 1% glutamax (Gibco, Thermo Fisher Scientific, Darmstadt, Germany), and 1% Anti/Anti (Gibco, Thermo Fisher Scientific, Darmstadt, Germany). Twenty micromolar HEPES buffer (Sigma-Aldrich, Darmstadt, Germany) was added to the RPMI-1640 medium. Tumor cells were cultivated in a humidified, 5% CO_2_ incubator.

### 4.2. Resistance Induction and Application of Docetaxel and Shikonin

DX-resistant sublines were established by continuous exposure to stepwise increasing drug concentrations as previously described [74]. The DX-resistant tumor cells were exposed to 12.5 nM DX (Sigma-Aldrich, Darmstadt, Germany) three times a week. Therapy-sensitive (parental) PCa cells served as controls. Shikonin (SHI) (Sigma-Aldrich, Darmstadt, Germany) was applied for 24, 48, or 72 h at a concentration of 0.1–1.5 μM. Controls (parental and DX-resistant) remained SHI-untreated. The IC50 (half-maximal inhibitory concentration) of SHI in parental and DX-resistant PCa cells was evaluated using the 72 h growth data at a concentration of 0.1–1.5 μM SHI. To evaluate possible toxic effects of DX and/or SHI, cell viability was determined parallel to experimentation by testing aliquoted cells with trypan blue (Sigma-Aldrich, Darmstadt, Germany). Only viable cells were used for growth and proliferation assays (see Section 4.3 and Section 4.4).

### 4.3. Tumor Cell Growth

Cell growth was assessed using 3-(4,5-dimethylthiazol- 2-yl)-2,5-diphenyltetrazolium bromide (MTT) dye. PCa cells (50 µL, 1 × 10^5^ cells/mL) were seeded into 96-well plates. After 24, 48, and 72 h, 10 µL MTT (0.5 mg/mL) (Sigma-Aldrich, Darmstadt, Germany) was added for 4 h. Cells were then lysed in 100 µL solubilization buffer containing 10% SDS in 0.01 M HCl. The plates were subsequently incubated overnight at 37 °C, 5% CO_2_. Absorbance at 570 nm was determined for each well using a multimode microplate-reader (Tecan, Spark 10 M, Crailsheim, Germany). After subtracting background absorbance and offsetting with a standard curve, results were expressed as mean cell number. To illustrate dose-response kinetics, the mean cell number after 24 h incubation was set to 100%. Each experiment was done in triplicate.

### 4.4. Proliferation

Cell proliferation was measured using a BrdU (bromodeoxyuridine/5-bromo-2′-deoxyuridine) cell proliferation enzyme-linked immunosorbent assay (ELISA) kit (Calbiochem/Merck Biosciences, Darmstadt, Germany). Tumor cells (50 µL, 1 × 10^5^ cells/mL), seeded into 96-well plates, were incubated with 20 µL BrdU-labeling solution per well for 24 h, and fixed and stained using anti-BrdU mAb according to the manufacturer’s protocol. Absorbance was measured at 450 nm using a multimode microplate-reader (Tecan, Spark 10 M, Crailsheim, Germany). Values were presented as percentage compared to untreated controls set to 100%.

### 4.5. Cell Cycle Phase Distribution

For cell cycle analysis, cell cultures were grown to sub-confluency. A total of 1 × 10^6^ cells was stained with propidium iodide (50 µg/mL) (Invitrogen, Thermo Fisher Scientific, Darmstadt, Germany) and then subjected to flow cytometry (Fortessa X20, BD Biosciences, Heidelberg, Germany). Ten-thousand events were collected from each sample. Data acquisition was carried out using DIVA software (BD Biosciences, Heidelberg, Germany), and cell cycle distribution was analyzed by ModFit LT 5.0 software (Verity Software House, Topsham, ME, USA). The number of cells in the G0/G1, S, or G2/M phases was expressed as a percentage. Untreated cells served as controls (dotted line; set to 100%).

### 4.6. Western Blot Analysis of Cell Cycle and Cell Death Regulating Proteins

To explore the expression and activity of cell cycle and cell death regulating proteins, Western blot analysis was performed. Tumor cell lysates (50 µg) were applied to 10% or 12% polyacrylamide gels and separated for 10 min at 80 V and for ~60–90 min at 120 V. The proteins were then transferred to nitrocellulose membranes (1 h, 100 V). After blocking with 10% non-fat dry milk for 1 h, the membranes were incubated overnight with the following primary antibodies directed against cell cycle proteins: p21 (Rabbit IgG, clone 12D1, dilution 1:1000, Cell Signaling, Frankfurt am Main, Germany), p27 (Mouse IgG_1_, clone 57/Kip1, dilution 1:500, BD Biosciences, Heidelberg, Germany), Cyclin A (Mouse IgG_1_, clone 25, dilution 1:500, BD Biosciences, Heidelberg, Germany), Cyclin B (Mouse IgG_1_, clone 18, dilution 1:1000, BD Biosciences, Heidelberg, Germany), CDK1 (Mouse IgG_1_, clone 2, dilution 1:2500, BD Biosciences, Heidelberg, Germany), CDK2 (Mouse IgG_2a_, clone 55, dilution 1:2500, BD Biosciences, Heidelberg, Germany), and pCDK2 (Rabbit, polyclonal antibody, dilution 1:1000, Cell Signaling, Frankfurt am Main, Germany).

To detect apoptosis- and necroptosis-related proteins, the following primary antibodies were used: Caspase 3 (Rabbit IgG, polyclonal antibody, dilution 1:1000), Caspase 8 (Rabbit IgG, clone D35G2, dilution 1:1000), PARP (Rabbit IgG, clone 46D11, dilution 1:1000), RIP1 (Rabbit IgG, clone D94C12, dilution 1:1000), pRIP1S166 (Rabbit IgG, clone D1L3S, dilution 1:1000), RIP3 (Rabbit IgG, clone E1Z1D, dilution 1:1000), pRIP3S227 (Rabbit IgG, clone D6W2T, dilution 1:1000), MLKL (Rabbit IgG, clone D2I6N, dilution 1:1000), and pMLKLS358 (Rabbit IgG, clone D6H3V, dilution 1:1000) (all Cell Signaling, Frankfurt am Main, Germany). HRP-conjugated rabbit-anti-mouse IgG or goat-anti-rabbit IgG served as secondary antibodies (IgG, both: dilution 1:1000, Dako, Glosturp, Denmark). The membranes were incubated with ECL detection reagent (AC2204, Azure Biosystems, Munich, Germany) to visualize proteins with a Sapphire Imager (Azure Biosystems, Munich, Germany). The exposure time was adapted to the signal intensity (device-specific maximum, >65,000 = oversaturated). Only images with a maximum band intensity of below 65,000 were used for evaluation. β-actin (clone AC-1, dilution 1:10,000, Sigma Aldrich, Taufkirchen, Germany) served as internal control for cell cycle regulating proteins. Cell death regulating proteins were normalized to total protein that was quantified by staining total protein from all membranes with Coomassie brilliant blue and measuring with a Sapphire Imager. AlphaView software (ProteinSimple, San Jose, CA, USA) was used for pixel density analysis of the protein bands. The ratio of protein intensity/β-actin intensity or whole protein intensity was calculated and expressed in percentage, related to untreated controls, set to 100%.

### 4.7. Cell Death

To investigate apoptotic and necrotic events the binding of Annexin V/propidium iodide (PI) in PC3, DU145 and LNCaP cells was quantified with the FITC-Annexin V Apoptosis Detection kit (BD Biosciences, Heidelberg, Germany). After washing tumor cells twice with PBS, 1 × 10^5^ cells were suspended in 500 µL of 1 × binding buffer and incubated with 5 µL Annexin V-FITC and (or) 5 µL PI in the dark for 15 min. Staining was measured by flow cytometer (Fortessa X20, BD Biosciences, Heidelberg, Germany). Ten-thousand events were collected from each sample. The percentage of apoptotic and necrotic cells in each quadrant was calculated using DIVA software (BD Biosciences, Heidelberg, Germany). Further analysis was done by FlowJo software (BD Biosciences, Heidelberg, Germany).

An L-Lactate dehydrogenase Cytotoxicity Assay Kit (Thermo Scientific, Waltham, MA, USA) was used to evaluate cell death/cytotoxicity of 22Rv1. Tumor cells (50 µL, 1 × 10^5^ cells/mL) were seeded into 96-well plates and treated for 48 h with 50 µL 0.5, 0.75, and 1.0 µM SHI. After incubation, 50 µL medium supernatant of treated cells was transferred to a new 96-well plate, mixed with a reaction solution for 30 min, and then stopped with 50 µL stop solution, according to the manufacturer’s protocol. Absorbance was measured at 490 nm using a multimode microplate-reader (Tecan, Spark 10 M, Crailsheim, Germany). Values were presented as a percentage compared to untreated controls.

Necroptotic effects were assessed using 3-(4,5-dimethylthiazol- 2-yl)-2,5-diphenyltetrazolium bromide (MTT) dye. To evaluate necroptosis, tumor cells were treated for 24 h and 48 h with 0.5, 0.8, and 1.0 µM SHI or SHI combined with 80 µM necrostatin-1 (Sigma-Aldrich, Darmstadt, Germany), a necroptosis inhibitor or with 20 µM zVAD (Selleckchem, München, Deutschland), a multi-caspase inhibitor. For more details, see “Tumor Cell Growth” (Section 4.3).

### 4.8. GSH-Assay

The GSH level was evaluated with the GSH-Glo™ Glutathione Assay (Promega Corporation, Madison, WI, USA). Five-thousand cells/well were seeded onto a 96-well plate and incubated for 24 h with 0.5 µM SHI. Experiments were performed according to the manufacturer’s protocol. Luminescence was measured using a multimode microplate-reader (Tecan, Spark 10 M, Tecan, Grödig, Austria).

### 4.9. Evaluation of Mitochondrial Respiration and Anaerobic Glycolytic Activity

Mitochondrial respiration (OCR = oxygen consumption rate) and anaerobic glycolytic activity (EACR = extracellular acidification rate) were assessed in real-time by the Seahorse XFp Extracellular Flux Analyzer using the Seahorse XF Cell Mito Stress Test Kit (both: Agilent Technologies, Waldbronn, Germany). The OCR is defined by multiple parameters, including basal respiration, ATP production-coupled respiration, maximal and reserve capacities, and non-mitochondrial respiration. Cells stained with CellTracker Green CMFDA (Thermo Fisher Scientific, Darmstadt, Germany) were plated at a density of 2 × 10^4^ cells/well and media was replaced with XF Assay media the following day, 1 h prior to the assay and incubated without CO_2_. Five measurements of OCR and ECAR were done at baseline and 30 measurements after SHI injection. Data were normalized to the mean fluorescent intensity of cells in the area of measurement using Wave 2.6.1 (Agilent Technologies, Waldbronn, Germany) desktop software.

### 4.10. Statistical Analysis

All experiments were performed at least three times. The evaluation and generation of mean values, the associated standard deviation, and normalization in percent were done by Microsoft Excel. Statistical significance was calculated with GraphPad Prism 7.0 (GraphPad Software Inc., San Diego, CA, USA): two-sided *t*-test (Western blot, apoptosis, cell cycle), one-way ANOVA test (BrdU), and two-way ANOVA test (MTT). Correction for multiple comparison was done using the conservative Bonferroni method. Error bars indicate standard deviation (SD). Differences were considered statistically significant at a *p*-value ≤ 0.05 with * = *p* ≤ 0.05, ** = *p* ≤ 0.01, *** = *p* ≤ 0.001.

## 5. Conclusions

SHI induced a time- and dose-dependent inhibition of tumor cell growth and proliferation in a panel of parental and DX-resistant PCa cells. SHI’s growth inhibitory effect was accompanied by necroptosis induction in all PCa cell lines, including the DX-resistant cell lines. Exposure to SHI triggered necroptosis by decreasing caspase 8 and increasing pRIP1 and pRIP3. Notably, in the more aggressive, androgen-insensitive PCa cells—PC3 and DU145, the strongest necroptotic effects were apparent. Furthermore, evidence is presented showing that SHI may reactivate the necroptotic action of DX in those cells. SHI also contributed to a cell type specific cell cycle arrest in the G2/M or S phase with corresponding modulations of the cell cycle regulating proteins. In regard to these findings, it is postulated that SHI could hold promise as a beneficial addition to the conventional treatment of advanced PCa. Further investigation is necessary to evaluate other possible in vitro antitumor effects of SHI and to verify these in vivo.

## Figures and Tables

**Figure 1 cancers-13-00882-f001:**
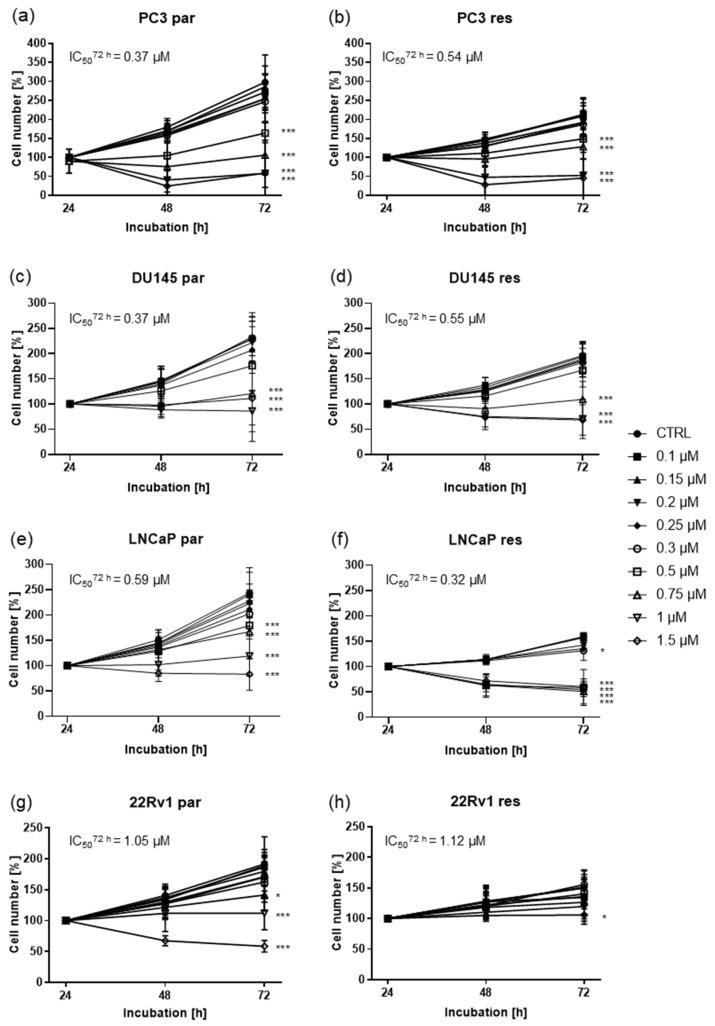
Tumor cell growth of parental (par) and DX-resistant (res) PC3 (**a**,**b**), DU145 (**c**,**d**), LNCaP (**e**,**f**), and 22Rv1 (**g**,**h**) cells after 24, 48, and 72 h exposure to shikonin (SHI) [0.1–1.5 µM]. Cell number set to 100% after 24 h incubation. The IC50 of SHI after 72 h treatment is specified. Error bars indicate standard deviation (SD). Significant difference to untreated control: * = *p* ≤ 0.05, *** = *p* ≤ 0.001. *n* = 5.

**Figure 2 cancers-13-00882-f002:**
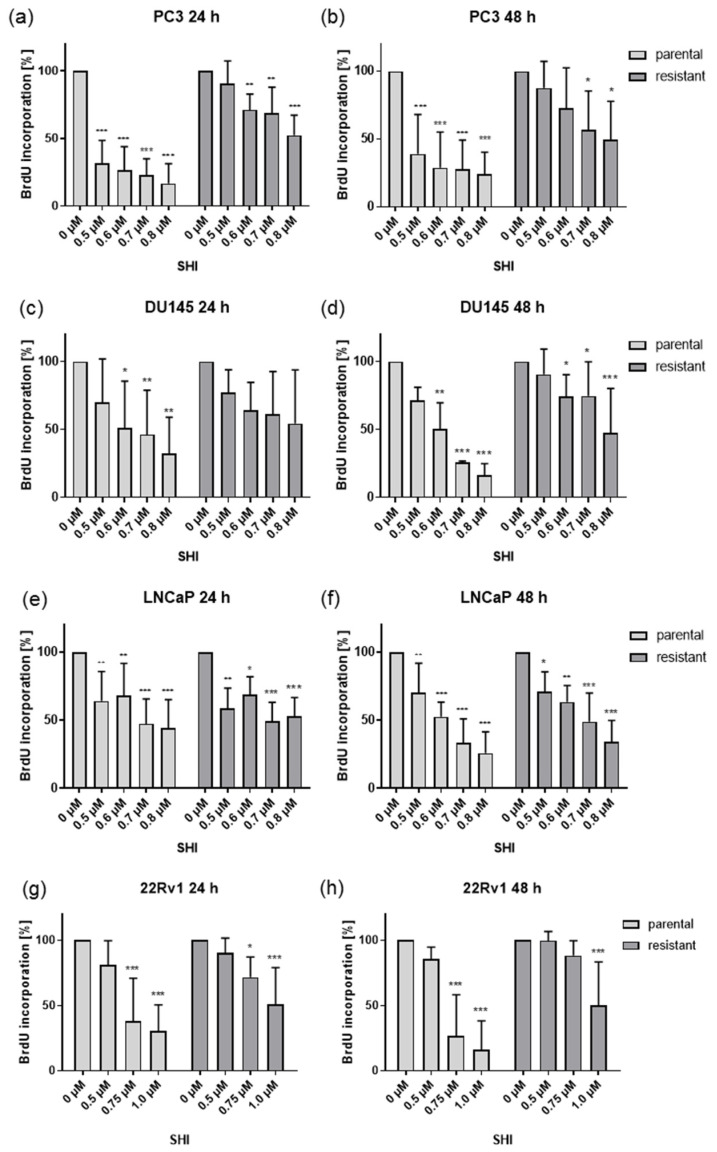
Tumor cell proliferation of parental and DX-resistant PC3 (**a**,**b**), DU145 (**c**,**d**), LNCaP (**e**,**f**), and 22Rv1 (**g**,**h**) PCa cells incubated for 24 h and 48 h with SHI [0.5–0.8 µM] (**a**–**f**) or [0.5–1.0] (**g**,**h**). Untreated controls were set to 100%. Error bars indicate standard deviation (SD). Significant difference to untreated control: * = *p* ≤ 0.05, ** = *p* ≤ 0.01, *** = *p* ≤ 0.001. *n* = 3.

**Figure 3 cancers-13-00882-f003:**
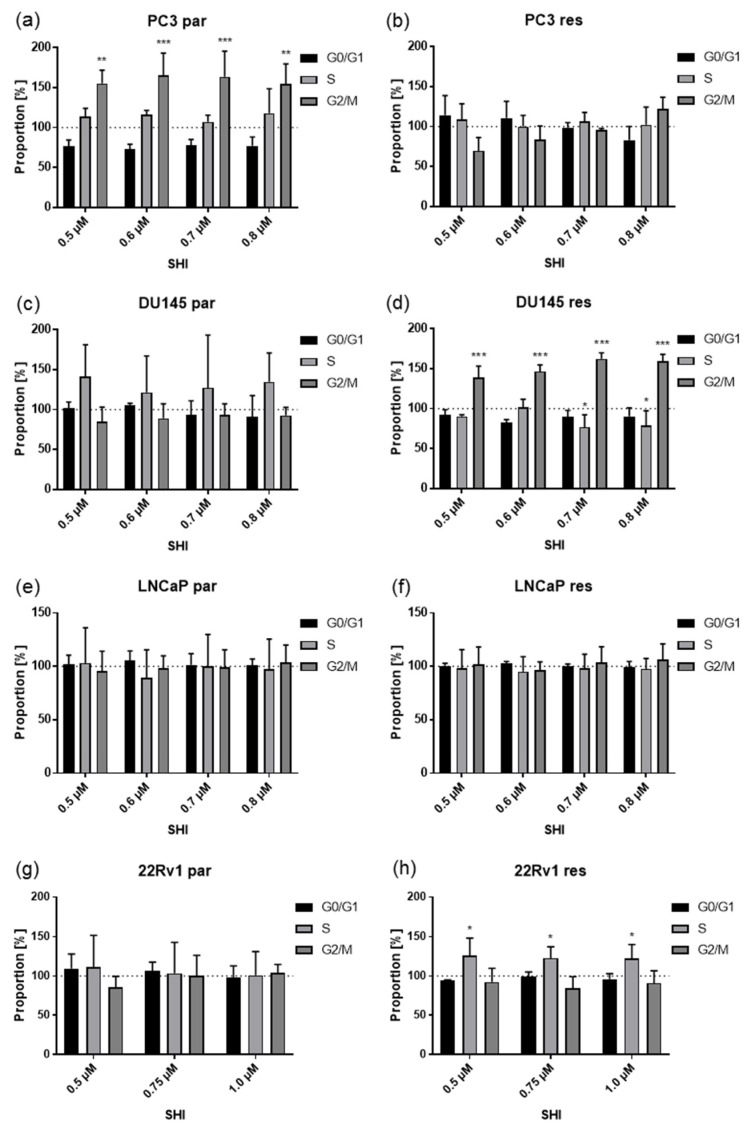
Distribution in the cell cycle phases: Proportion of parental (par) and DX-resistant (res) PCa cells, PC3 (**a**,**b**), DU145 (**c**,**d**), LNCaP (**e**,**f**), and 22Rv1 (**g**,**h**), in the G0/G1, S, and G2/M phases after 48 h treatment with SHI [0.5–0.8 µM] (**a**–**f**) and SHI [0.5–1.0 µM] (**g**,**h**). Untreated cells served as controls (dotted line; set to 100%). Error bars indicate standard deviation (SD). Significant difference to untreated control: * = *p* ≤ 0.05, ** = *p* ≤ 0.01, *** = *p* ≤ 0.001. *n* = 3.

**Figure 4 cancers-13-00882-f004:**
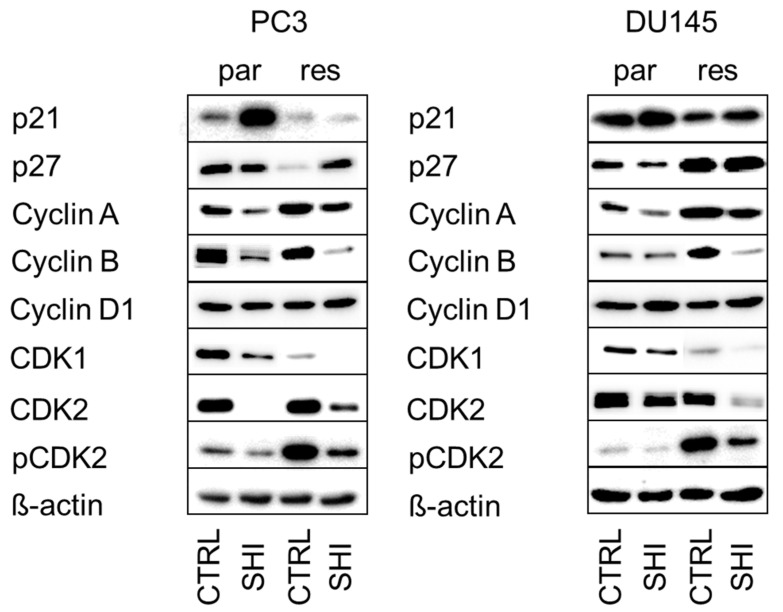
Protein expression profile of cell cycle regulating proteins: Representative Western blot images of cell cycle regulating proteins in parental (par) and DX-resistant (res) PC3 (left panel) and DU145 (right panel) cells after 48 h exposure to SHI [0.5 µM].

**Figure 5 cancers-13-00882-f005:**
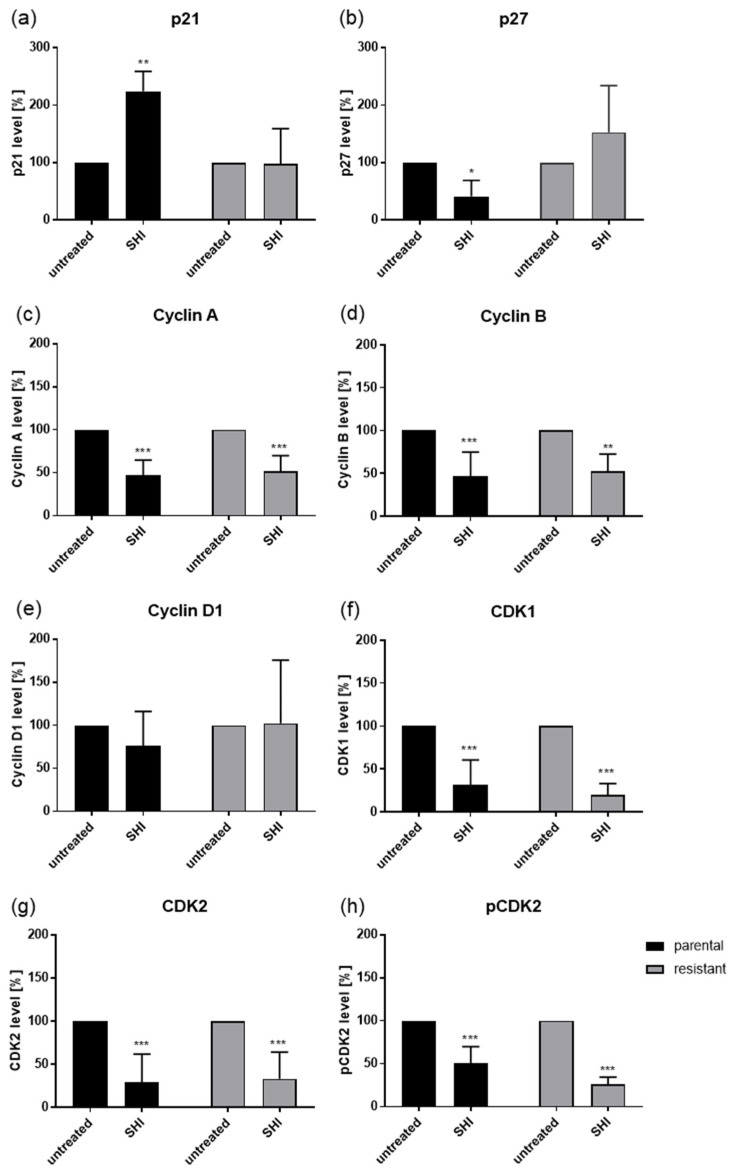
Protein expression profile of cell cycle regulating proteins: Pixel density analysis (Western blot) of the cell cycle regulating proteins p21 (**a**), p27 (**b**), Cyclin A (**c**), Cyclin B (**d**), Cyclin D1 (**e**), CDK1 (**f**), CDK2 (**g**), and pCDK2 (**h**) in parental and DX-resistant PC3 cells after 48 h exposure to SHI [0.5 µM], compared to untreated controls (set to 100%). Each protein analysis was accompanied and normalized by a housekeeping protein. Error bars indicate standard deviation (SD). Significant difference to untreated control: * = *p* ≤ 0.05, ** = *p* ≤ 0.01, *** = *p* ≤ 0.001. *n* = 3. For detailed information regarding the Western blots, see Appendix A.

**Figure 6 cancers-13-00882-f006:**
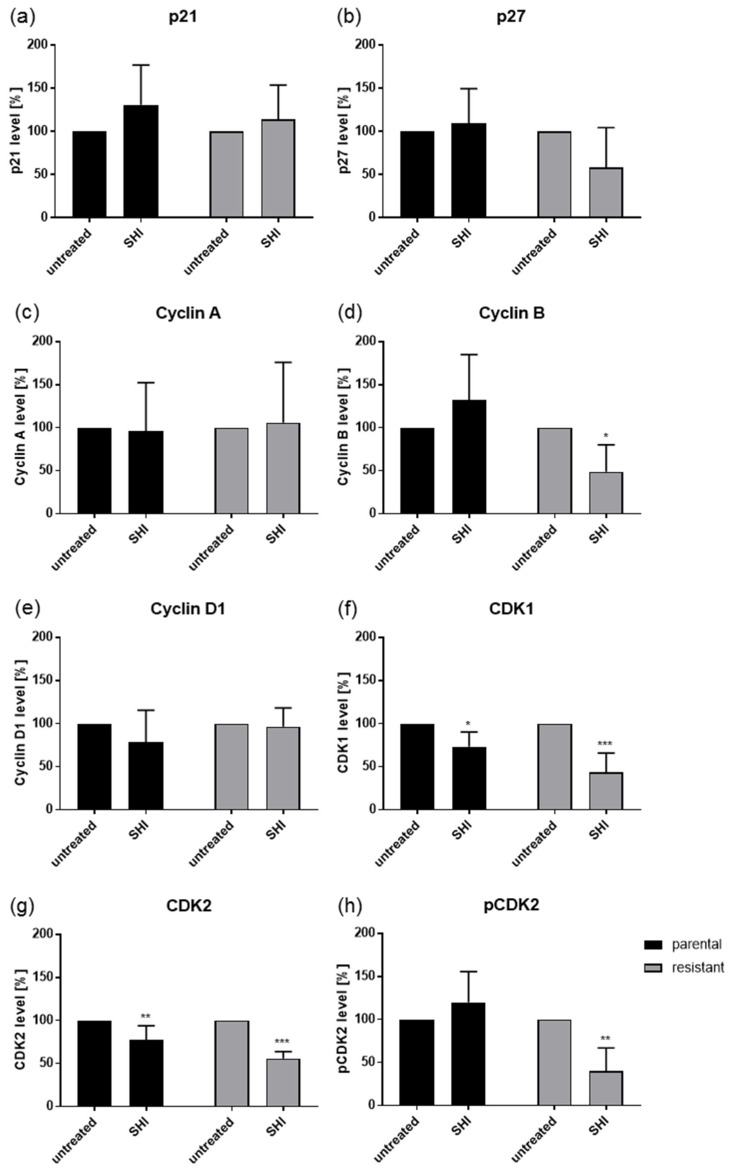
Protein expression profile of cell cycle regulating proteins: Pixel density analysis (Western blot) of the cell cycle regulating proteins p21 (**a**), p27 (**b**), Cyclin A (**c**), Cyclin B (**d**), Cyclin D1 (**e**), CDK1 (**f**), CDK2 (**g**), and pCDK2 (**h**) in parental and DX-resistant DU145 cells after 48 h exposure to SHI [0.5 µM], compared to untreated controls (set to 100%). Each protein analysis was accompanied and normalized by a housekeeping protein. Error bars indicate standard deviation (SD). Significant difference to untreated control: * = *p* ≤ 0.05, ** = *p* ≤ 0.01, *** = *p* ≤ 0.001. *n* = 3. For detailed information regarding the Western blots, see Appendix A.

**Figure 7 cancers-13-00882-f007:**
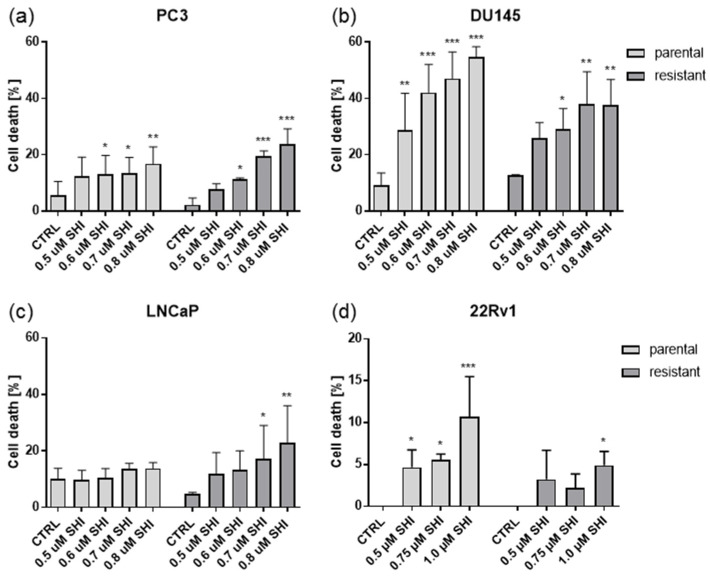
Cell death events: Percent of cell death in parental and DX-resistant PC3 (**a**), DU145 (**b**), LNCaP (**c**), and 22Rv1 (**d**) cells treated for 48 h with 0.5–0.8 µM SHI (**a**–**c**) or 0.5–1.0 µM SHI (**d**), compared to the untreated controls. Error bars indicate standard deviation (SD). Significant difference to untreated control: * = *p* ≤ 0.05, ** = *p* ≤ 0.01, *** = *p* ≤ 0.001. *n* = 3.

**Figure 8 cancers-13-00882-f008:**
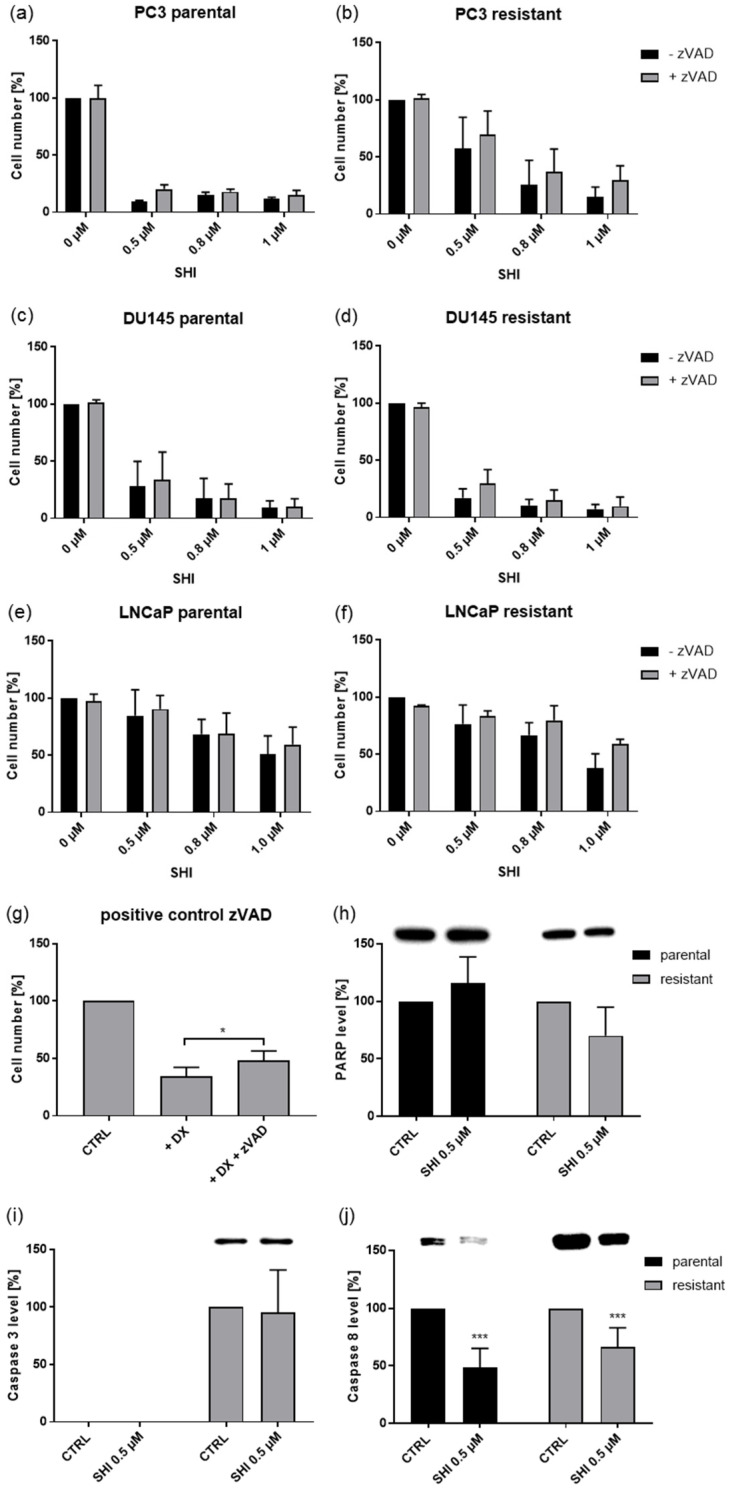
Caspase-dependent cell death: Cell growth of parental and DX-resistant PC3 and DU145 cells treated for 48 h with SHI [0.5, 0.8, 1.0 µM] and the multi-caspase inhibitor zVAD [20 µM] (**a**–**f**). SHI mono-treated and untreated (set to 100%) cells served as controls. Parental DU145 cells treated with 12.5 nM DX alone or in combination with 20 µM zVAD were used as a positive control to confirm zVAD activity (**g**). Error bars indicate standard deviation (SD). Significant difference compared to untreated controls, except for asterisk brackets indicating a significant difference between untreated and zVAD treated cells: * = *p* ≤ 0.05. *n* = 3 (**a**–**g**). Protein expression of PARP (**h**), caspase 3 (**i**), and caspase 8 (**j**) in parental and DX-resistant DU145 cells after 24 h exposure to 0.5 µM SHI: Representative Western blot images and pixel density analysis. Protein analysis was accompanied and normalized by a total protein control. Untreated cells served as controls (set to 100%). Error bars indicate standard deviation (SD). Significant difference to untreated control: *** = *p* ≤ 0.001. *n* = 3. For detailed information regarding the Western blots, see Appendix A.

**Figure 9 cancers-13-00882-f009:**
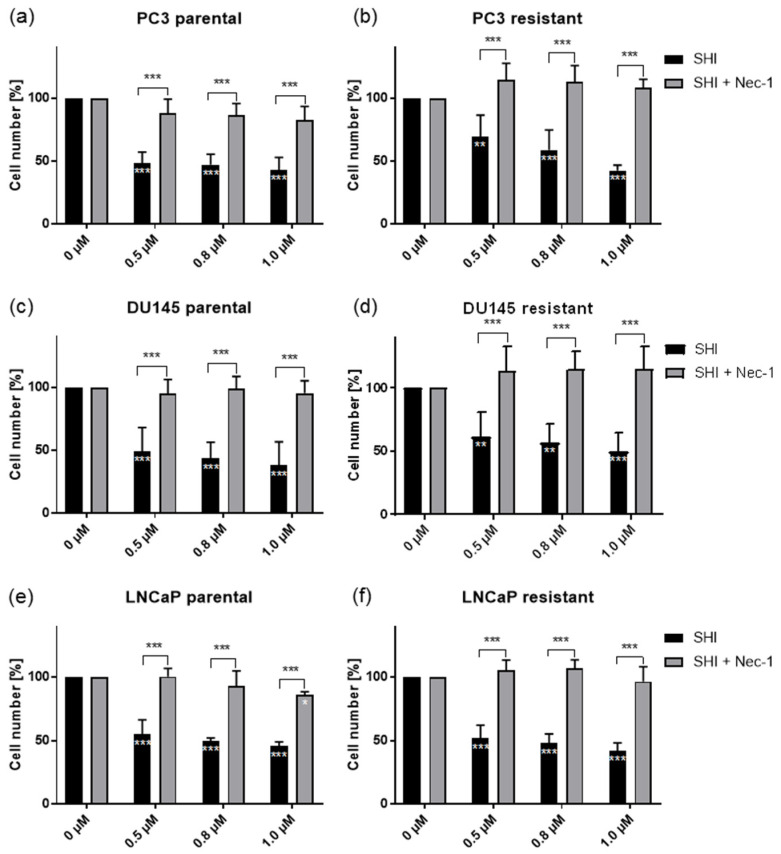
Necroptosis in PCa cells: Necroptosis induction of parental and DX-resistant PC3 (**a**,**b**), DU145 (**c**,**d**) and LNCaP (**e**,**f**) cells treated for 24 h with 0.5, 0.8, and 1.0 µM SHI and 80 µM necrostatin-1 (Nec-1). SHI mono-treated and untreated (set to 100%) cells served as controls. Error bars indicate standard deviation (SD). Significant difference, compared to untreated controls, except for asterisk brackets indicating a significant difference between Nec-1 untreated and treated cells: *** = *p* ≤ 0.001. *n* = 5.

**Figure 10 cancers-13-00882-f010:**
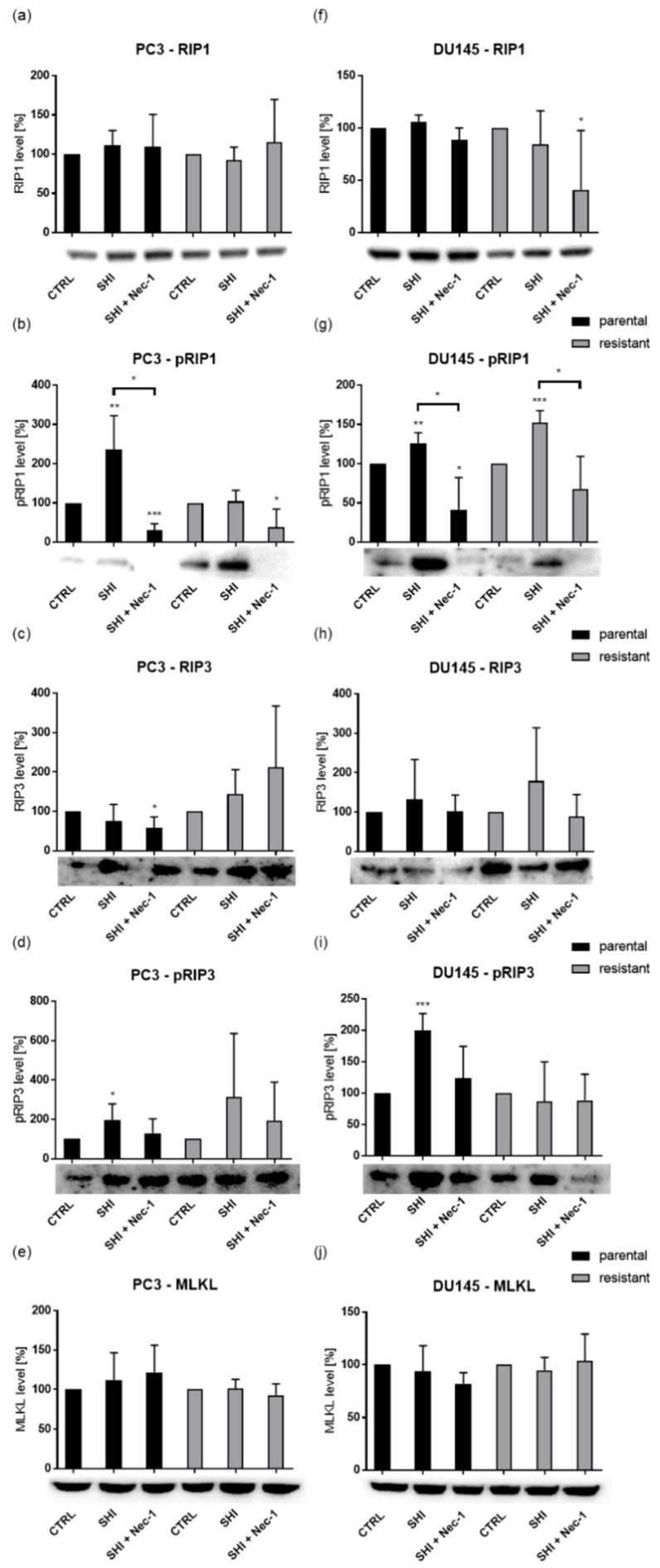
Expression and activity of necroptosis markers in PCa cells: Protein expression of RIP1 (**a**,**f**), pRIP1 (**b**,**g**), RIP3 (**c**,**h**), pRIP3 (**d**,**i**), and MLKL (**e**,**j**) in parental and DX-resistant PC3 (**a**–**e**) and DU145 (**f**–**j**) cells after 12 h exposure to 0.5 µM SHI and 80 µM necrostatin-1 (Nec-1). Representative Western blot images and pixel density analysis. Protein analysis was accompanied and normalized by a total protein control. Untreated cells served as controls (set to 100%). Error bars indicate standard deviation (SD). Significant difference to untreated control: * = *p* ≤ 0.05, ** = *p* ≤ 0.01, *** = *p* ≤ 0.001. *n* = 3. For detailed information regarding the Western blots, see Appendix A.

**Figure 11 cancers-13-00882-f011:**
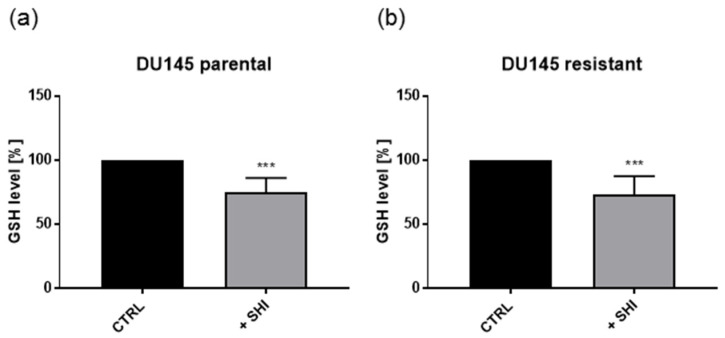
GSH content in DU145 cells: GSH was measured after 24 h exposure to 0.5 µM SHI. Untreated cells served as controls (set to 100%). Error bars indicate standard deviation (SD). Significant difference to untreated control: *** = *p* ≤ 0.001. *n* = 6.

## Data Availability

The data presented in this study are available in this article and Appendix A.

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
