# Peer review of "Shikonin Reduces Growth of Docetaxel-Resistant Prostate Cancer Cells Mainly through Necroptosis"

_cancers, 2021, doi:10.3390/cancers13040882_

Round 1
Reviewer 1 Report
The author used four parental and docetaxel-resistant PCa cell lines to illustrate that the possible mechanism of shikonin is to inhibit the growth of prostate cancer cells via necroptosis.
The author need to provide more evidence and explanation to help the integrity of the paper is necessary. Some suggested verifications and modifications are as follows:
- Shikonin suppressed the PCa growth and causing cell cycle arrest at G2/M and S phases. I’ll suggest the author provide the same dose of shikonin check in normal prostate epithelial cells such as RWPE1 to confirm the shikonin specificity without affected the normal prostate epithelium growth.
- Insufficient evidence of PCA necroptosis after shikonin treatment; the key targets of necroptosis such as phospho-RIPK1, phospho-RIPK3, and MLKL should be provided by IF, RT-qPCR, or western blot.
- In result 2.6, Figure S7 2B significantly enhanced the OCR of DU145 or DU145_DX and reduced ECAR when treated with shikonin. It is inconsistent with the content description and needs correction. In other words, if the authors can prove that shikonin can reduce the glycolysis of cancer cells by inhibiting PKM2 and converting it into the OXPHOS pathway, it will provide a possible mechanism for improving the novelty of this article.
- If possible, in vivo animal PCa xenograft study should be carried out with shikonin treatment, as well as validate the necroptosis marker in the animal tissue section at the endpoint.
Author Response
Comment 1: Shikonin suppressed the PCa growth and causing cell cycle arrest at G2/M and S phases. I’ll suggest the author provide the same dose of shikonin check in normal prostate epithelial cells such as RWPE1 to confirm the shikonin specificity without affected the normal prostate epithelium growth.
Our answer: The aim of our investigation was to evaluate whether shikonin (SHI) has an impact on progressive growth, not only in therapy-sensitive prostate cancer cells, but more importantly, therapy-resistant prostate cancer cells. Furthermore, it was our intent to give first insight into possible underlying functional and molecular mechanisms connected with SHI. We agree with you that further investigation in regard to possible toxic effects is important. Plans have been made to evaluate this aspect in future in vivo experiments, where toxic effects will be measurable not only for the prostate, but in general. Some work has already been carried out in regard to toxicity. In an osteosarcoma murine model intraperitoneal injection of 2 mg/kg SHI had no impact on the animals’ general conditions, e.g. alertness and physical activity (Fu et al., BMC Cancer 2013, doi: 10.1186/1471-2407-13-580). Also, no histopathological changes of the liver and kidney were observed after SHI treatment in a murine melanoma xenograft model (Lee et al., Biosci Rep 2021, doi: 10.1042/BSR20203834). In a colon cancer PDX model SHI alleviated liver and kidney dysfunction, and bodyweight remained stable during treatment [1 and 2 mg SHI/kg] (Chen et al., Sci Rep 2020, doi: 10.1038/s41598-020-71116-5).
Regarding in vivo experiments, please also see our answer to comment 4.
Comment 2: Insufficient evidence of PCA necroptosis after shikonin treatment; the key targets of necroptosis such as phospho-RIPK1, phospho-RIPK3, and MLKL should be provided by IF, RT-qPCR, or western blot.
Our answer: Indeed, further in-depth studies in regard to the underlying molecular mechanism of necroptosis would be interesting. However, the main focus of this investigation was to evaluate whether SHI has an impact on the progressive growth of parental and docetaxel (DX)-resistant prostate carcinoma cells. In fact, SHI significantly inhibited growth and proliferation, partially due to cell cycle arrest and down-regulation of cell cycle regulating proteins, but predominantly through inducing cell death. Necrotic effects were excluded. Interestingly, treatment with zVAD, a multi-caspase inhibitor (also called apoptosis inhibitor), revealed that cell death was caspase-independent. Further analysis after SHI treatment on the protein level identified no effect on PARP or caspase 3, excluding DNA-damage repair or intrinsic apoptosis. Down-regulation of caspase 8, an initiator of extrinsic apoptosis and a necroptosis inhibitor was also apparent, further indicating a caspase-independent cell death induction, such as necroptosis. We pursued that evidence by investigating the impact of the necroptosis inhibitor, necrostatin-1 and indeed found a significant reversal of SHI’s anti-growth properties with its application. This corroborated our hypothesis that SHI induces necroptosis. For further confirmation, we representatively evaluated the expression of RIP1, part of the necrosome complex, involved in necroptosis induction, after treating the parental and docetaxel-resistant prostate cancer cells with SHI. In good accordance with our previous data, RIP1 expression was significantly enhanced in the parental tumor cells after SHI exposure, further supporting that SHI induces necroptosis. We have now added new data regarding the glutathione (GSH)-content in the cells.
The following has been added:
Results section lines 275-276 and Figure 10:
“In addition, administration of 0.5 µM SHI resulted in a significant decrease of the GSH-content in parental and DX-resistant DU145 cells (Figure 10), indicating ROS generation.
Figure 10. GSH-content in DU145 cells: GSH was measured after 24 h exposure to 0.5 µM SHI. Untreated cells served as controls (set to 100%). Error bars indicate standard deviation (SD). Significant difference to untreated control: *** = p ≤ .001. n = 6.”
Discussion section lines 387-392: “The GSH-content was also significantly diminished after SHI application in the parental and DX-resistant PCa cells, indicating ROS generation. Necroptosis induction has also been shown to be accompanied by increased ROS levels in nasopharyngeal carcinoma cells [46]. SHI-induced GSH depletion and intracellular ROS increase in tumor cells has been demonstrated to be RIP1- and RIP3-mediated [67,69], as well, further confirming that SHI induces necroptosis, as observed in the current investigation.”
Materials and Methods section lines 514-519: “4.8. GSH-Assay
The GSH level was evaluated with the GSH-Glo™ Glutathione Assay (Promega Corporation, Madison, Wisconsin, USA). Five thousand cells/well were seeded onto a 96-well-plate and incubated for 24 h with 0.5 µM SHI. Experiments were performed according to the manufacturer’s protocol. Luminescence was measured using a multi-mode microplate-reader (Tecan, Spark 10 M, Tecan, Grödig, Austria).”
Notably, the GSH-content significantly diminished after SHI application in the parental and DX-resistant PCa cells, indicating reactive oxygen species (ROS) generation. SHI-induced GSH depletion and intracellular ROS increase in tumor cells has been shown to be RIP1- and RIP3-mediated further confirming necroptosis induction found in the current investigation.
We were thus able to exclude extrinsic and intrinsic apoptosis induction by caspase 3 and 8 and further confirmed a possible necroptosis induction with the reduction of caspase 8 after SHI treatment. This was corroborated by use of the necroptosis inhibitor necrostatin-1, elevated RIP1 expression, and reduction of the GSH-content. Certainly, further components of the necroptosis pathway should be evaluated and more in-depth studies of the underlying mechanism should be done in the future. But since necroptosis is a complex subject, relegation to a future investigation would be more appropriate. In the current study we show, for the first time, that SHI inhibits progressive growth behavior, not only of parental, but more importantly of DX-resistant prostate cancer cells, and that necroptosis is involved.
Comment 3: In result 2.6, Figure S7 2B significantly enhanced the OCR of DU145 or DU145_DX and reduced ECAR when treated with shikonin. It is inconsistent with the content description and needs correction. In other words, if the authors can prove that shikonin can reduce the glycolysis of cancer cells by inhibiting PKM2 and converting it into the OXPHOS pathway, it will provide a possible mechanism for improving the novelty of this article.
Our answer: Indeed, there was a significant short-term increase in OCR in the DU145 cells, partially accompanied by decreased glycolysis immediately after treatment with SHI. However, this effect was very short, directly diminishing after initiation and was no longer detectable after ~2 hours. The short-lasting metabolic shift towards mitochondrial respiration might indicate a temporary avoidance of apoptosis induction, as has previously been proposed (Shimada et al., Arch Biochem Biophys 2018, doi: 10.1016/j.abb.2017.12.008). Necroptosis induction by SHI has been postulated to overcome apoptosis resistance (Wang et al., Int J Biol Sci 2018, doi: 10.7150/ijbs.27854; Shahsavari et al., Asian Pac J Cancer Prev. 2015, doi: 10.7314/apjcp). Indeed, SHI-induced necroptosis prevented tumor escape and resulted in significant growth inhibition of our DU145 cells. The growth inhibition was comparable to that in PC3 cells, which did not show a short-term increase in OCR. Thus, the temporary OCR elevation in the DU145 cells did not facilitate tumor cell survival. However, we now included this short-term effect of SHI on mitochondrial respiration in the revised version of the manuscript and have corrected the text as follows:
Results section: “SHI has been shown to directly influence mitochondrial activity [38] and serve as a specific pyruvate kinase M2 inhibitor [24]. Inhibiting this enzyme with SHI could therefore directly influence tumor cell metabolism. Basal oxygen consumption rate and extracellular acidification rate were comparable in parental and DX-resistant PCa cells. However, treatment with SHI resulted only in temporarily elevated mitochondrial respiration and decreased aerobic glycolysis in the DU145 cells, indicating transient enhanced oxidative phosphorylation in response to a stress stimulus (Figure S7).” (lines 282-289).
Discussion section: “SHI has also been described to directly or indirectly influence the metabolism of cancer cells [38,70]. In the current investigation SHI only induced a significant short-term increase in OCR in the DU145 cells, partially accompanied by decreased glycolysis. This short-lasting metabolic shift towards mitochondrial respiration might indicate a temporary avoidance of apoptosis induction, as has previously been hypothesized [71]. Necroptosis induction by SHI has been postulated to overcome apoptosis resistance [34,72]. Indeed, SHI-induced necroptosis prevented tumor escape, resulting in significant growth inhibition of the PCa cells” (lines 393-401).
Comment 4: If possible, in vivo animal PCa xenograft study should be carried out with shikonin treatment, as well as validate the necroptosis marker in the animal tissue section at the endpoint.
Our answer: In vivo studies are planned in collaboration, and will be performed as soon as financial support is granted. When doing in vivo experiments we want to validate as many of the current in vitro findings as possible. This entails careful planning so that progressive growth behavior, metastasis formation and toxicity can be evaluated in the least possible number of animals, following the ethic 3R (replace, reduce and refine) policy. Thus, speedy in vivo experiments exceed the bounds of this investigation.

Reviewer 2 Report
This is an interesting article that establishes molecular response pathways to Shikonin treatment in prostate cancer cell lines. This has not been done before, in such a careful way although, the compound is known for some time (this diminishes some novelty of the paper).
Overall impact is also diminished by lack of in vivo, xenograft data, for the cell lines used in the current manuscript. However, the paper careful analysis of cell lines and sensitivity data will be very useful for such studies and could serve as useful guide.
The manuscript could be improved, however, by including some additional experimental data. Does anti-AR treatment (enzalutamide or another anti-AR agent) synergize with Shikonin? These data will significantly enhance the paper, particularly, in the era of the widespread clinical use of anti-AR agents.
Minor comments are of editorial nature: Figure legends have some repetitive descriptive information related to statistical analysis (which is a strength of the paper!); this could be presented in the method section in a more general form. Discussion should be shortened. The schematic diagram depicting the Shikonin mechanisms and pathways could provide nice summary for the drug action.
Author Response
Comment 1: The manuscript could be improved, however, by including some additional experimental data. Does anti-AR treatment (enzalutamide or another anti-AR agent) synergize with Shikonin? These data will significantly enhance the paper, particularly, in the era of the widespread clinical use of anti-AR agents.
Our answer: We agree that combined treatment with shikonin (SHI) and enzalutamide or abiraterone in the parental and docetaxel (DX)-resistant tumor cells is of interest. Since chemotherapy with DX is another treatment option in the therapy of prostate cancer and therapy resistance is a main problem in the advanced stage, the focus of our current investigation was to evaluate the impact of SHI on the progressive growth not only of parental, but more importantly, of resistant prostate cancer cells. In this case we used DX-resistant cells. Adding an additional compound would mean a new approach, besides making it necessary to repeat the whole evaluation of functional and molecular mechanisms. It would also necessitate further examination of the AR pathways. This would really go beyond the scope of this investigation but is certainly a worthwhile avenue, considering the general problem of chemoresistance.
Comment 2: Minor comments are of editorial nature: Figure legends have some repetitive descriptive information related to statistical analysis (which is a strength of the paper!); this could be presented in the method section in a more general form.
Our answer: Thank you for the recommendation. Accordingly, we have added further information to the Materials and Methods section.
Under “4.5. Cell Cycle Phase Distribution” information about the controls at the end of the paragraph has been amended: “The number of cells in the G0/G1, S, or G2/M phases was expressed as a percentage. Untreated cells served as controls (dotted line; set to 100%).” (lines 456-458).
Moreover, we added information about the standard deviation and different asterisks with the corresponding p-values under “4.10 Statistical Analysis”: “Error bars indicate standard deviation (SD). Differences were considered statistically significant at a p-value ≤ .05 with * = p ≤ .05, ** = p ≤ .01, *** = p ≤ .001.” (lines 538-539).
Comment 3: Discussion should be shortened.
Our answer: After repeated rigorous review we shortened the Discussion section as follows. The affected paragraphs now read:
“Also, in pancreatic, lung [25], breast [31], gastric cancer cells [26], and melanoma [49] the administration of SHI resulted in cell cycle arrest in the G2/M phase.” (lines 306-307).
“Consistent with the cell cycle data, DX-resistant DU145 cells showed a stronger down-regulation of Cyclin B, CDK1, and CDK2 by SHI, compared to the parental cells. In line with the current data…” (lines 314-317).
“Since Cyclin B in complex with CDK1, mediates the transition from the G2 to M phase [52] the down-regulation of Cyclin B by SHI may prevent the transition from the G2 to M phase.” (lines 322-325).
“In leukemia cells SHI led to apoptosis, postulated to be associated with the inhibition of c-Myc [54].” (lines 335-336).
“Since SHI inhibited AR [48], which prevents cell death processes through the tumor necrosis factor-α (TNF-α) [64], this inhibition might play a crucial role in the necroptosis induction in LNCaP.” (lines 374-376).

Round 2
Reviewer 1 Report
The author’s response through cited relevant literature to support his research findings. However, most of the suggested experiments were not provided, only shown the shikonin reduces PCa growth phenomenon via necroptosis, but the molecular mechanism path was not clear. As suggested in comment 2, the key targets of necroptosis such as phospho-RIPK1, phospho-RIPK3, and MLKL should be provided by IF, RT-qPCR, or western blot.
Author Response
Comment 1: The author’s response through cited relevant literature to support his research findings. However, most of the suggested experiments were not provided, only shown the shikonin reduces PCa growth phenomenon via necroptosis, but the molecular mechanism path was not clear. As suggested in comment 2, the key targets of necroptosis such as phospho-RIPK1, phospho-RIPK3, and MLKL should be provided by IF, RT-qPCR, or western blot.
Our answer: As suggested, we performed Western blot analysis of pRIP1, pRIP3, and pMLKL to show the underlying mechanism of necroptosis inducted by SHI not only in parental and DX-resistant DU145 but also PC3 (which were not included previously) cells. Furthermore, we investigated the content of total RIP1, RIP3, and MLKL in the PCa cells. In preliminary tests, we first evaluated different antibodies. Besides, we adjusted the incubation time with SHI (and necrostatin-1) to 12 h, with the intent to detect as much phosphorylation effects of the three proteins as possible. However, pMLKL was not detectable (only once very weak), most probably because MLKL is the last of the proteins in the necrosome complex and thus its phosphorylation might appear later.
Accordingly, the following passages have been added or changed in the manuscript:
Abstract section (lines 44-45):
“This was shown by enhanced pRIP1 and pRIP3 expression and returned growth if applying the necroptosis inhibitor necrostatin-1.”
Results section (lines 248-301): Figure 9 has been changed and a new Figure (Figure 10) has been added, the text has been amended accordingly:
“2.5. Shikonin induced necroptotic effects
Necroptosis is a caspase-independent cell death and necrostatin-1 inhibits the activity of RIP1 and blocks the necroptosis pathway. Since SHI induced necroptosis, in various tumors [28,30,32,34], necrostatin-1 was applied to determine whether SHI also has an impact on PCa tumor cell growth. SHI application significantly reduced growth in all cell lines (Figure 9a – f). Combined administration of 0.5 - 1.0 µM SHI and 80 µM necrostatin-1 resulted in a reversal of SHI’s anti-growth effect in all parental and DX-resistant PCa cell lines, leading to cell growth comparable to the untreated controls (Figure 9a - f).
Figure 9. Necroptosis in PCa cells: Necroptosis induction of parental and DX-resistant PCa cells treated for 24 h with 0.5, 0.8, 1.0 µM SHI and 80 µM necrostatin-1 (Nec-1). SHI mono-treated and untreated (set to 100%) cells served as controls. Error bars indicate standard deviation (SD). Significant difference, compared to untreated controls, except for asterisk brackets indicating a significant difference between Nec-1 untreated and treated cells: *** = p ≤ .001. n = 5.
Representative for the tested PCa cell lines, PC3 and DU145 cells showed an increase in pRIP1 and/or pRIP3 activation after exposure to SHI (Figure 10b, d, g, i and Figure S6b, d, g, i). In parental PC3 pRIP1 and pRIP3 were significantly activated by SHI, whereas additional application of necrostatin-1 reversed this activation (Figure 10b and Figure S6b). DX-resistant PC3 cells revealed no effect on pRIP1 after exposure to SHI but displayed by tendency an elevation of pRIP3, compared to the SHI-untreated controls (Figure 10d and Figure S6d). Again, combined treatment with SHI and necrostatin-1 counteracted this activation and led to a decrease of pRIP1 and pRIP3, compared to the SHI-treated cells. In the DU145 cells both parental and stronger DX-resistant DU145 cells showed a significant up-regulation of pRIP1 by SHI (Figure 10g and Figure S6g). Addition of necrostatin-1 to SHI in parental and DX-resistant DU145 cells significantly abolished RIP1 phosphorylation. pRIP3 was also significantly amplified after SHI application in parental DU145 cells (Figure 10i and Figure S6i). As before, phosphorylation was abrogated by combining SHI with necrostatin-1. In contrast, the expression of total RIP1, RIP3 and MLKL was not significantly affected by SHI (Figure 10a, c, e, f, h, j and Figure S6a, c, e, f, h, j) and pMLKL was not detectable in the PC3 and DU145 cells. Combined application to SHI and necrostatin-1 significantly reduced the total amount of RIP3 in parental PC3 (Figure 10c and Figure S6c) and of RIP1 in DX-resistant DU145 (Figure 10f and Figure S6f).
Figure 10. Expression and activity of necroptosis markers in PCa cells: Protein expression of RIP1 (a, f), pRIP1 (b, g), RIP3 (c, h), pRIP3 (d, i), and MLKL (e, j) in parental and DX-resistant PC3 (a-e) and DU145 (f-j) cells after 12 h exposure to 0.5 µM SHI and 80 µM necrostatin-1 (Nec-1). Representative Western blot images and pixel density analysis. Protein analysis was accompanied and normalized by a total protein control. Untreated cells served as controls (set to 100%). Error bars indicate standard deviation (SD). Significant difference to untreated control: * = p ≤ .05, ** = p ≤ .01, *** = p ≤ .001. n = 3. For detailed information regarding the Western blots, see Figure S6a-j.
…”
The Discussion section now reads (lines 399-440):
“RIP1 and RIP3 are critical proteins involved in necroptosis induction [32]. Consistent with this, S166 phosphorylation of RIP1, indicating induction of necroptotic signaling, was evident in parental PC3 and DU145 cells, as well as in DX-resistant DU145 and by tendency in PC3 cells after SHI treatment. Also in gastric cancer cell lines [28], glioma [62], and osteosarcoma cells [55] administration of SHI resulted in a significant increase of RIP1 and RIP3. In the parental PCa cells pRIP1 facilitated phosphorylation of RIP3, downstream effector of the necroptotic complex. Elevated phosphorylation of RIP1 and RIP3 was reversed when necrostatin-1 was added. pMLKL is another component of the necrosome complex downstream of RIP3 [63]. MLKL is recruited and phosphorylated by pRIP3, the next step in initiation of necroptosis. In the PCa cells no pMLKL was detectable after exposure to SHI, as a downstream target probably occurring after the chosen 12 h incubation. The application period might also explain why the parental PCa cells showed stronger effects in the phosphorylation of RIP3 than the DX-resistant cells, although necroptotic effects were more pronounced - but at a later time point. However, after 12 h SHI treatment pRIP1 and pRIP3 were upregulated in the parental and by tendency in the DX-resistant PCa cells, further confirming the postulated functional role of SHI in necroptosis initialization.
The GSH-content was also significantly diminished after SHI application in the parental and DX-resistant PCa cells, indicating ROS generation. Necroptosis induction has also been shown to be accompanied by increased ROS levels in nasopharyngeal carcinoma cells [46]. SHI-induced GSH depletion and intracellular ROS increase in tumor cells has been demonstrated to be RIP1- and RIP3-mediated [62,64] as well, further confirming that SHI induces necroptosis, as observed in the current investigation.
Hence, the measured “apoptotic” effects mainly seem to be due to necroptosis. However, the SHI treatment of LNCaP cells revealed only marginal apoptosis, indicating another mechanism. In contrast to the other tested PCa cells, LNCaP cells are androgen receptor (AR)-positive and androgen-sensitive [65,66]. Since SHI inhibited AR [48], which prevents cell death processes through the tumor necrosis factor-α (TNF-α) [67], this inhibition might play a crucial role in the necroptosis induction in LNCaP. Notably, TNF-α is involved in necroptotic processes [68,69]. However, the role of TNF-α in LNCaP cells requires further investigation.
…”
In the Material and Methods section we added (lines 508, 522-530):
“4.6. Western Blot Analysis of Cell Cycle and Cell Death Regulating Proteins
…To detect apoptosis and necroptosis related proteins the following primary antibodies were used: Caspase 3 (Rabbit IgG, polyclonal antibody, dilution 1:1000), Caspase 8 (Rabbit IgG, clone D35G2, dilution 1:1000), PARP (Rabbit IgG, clone 46D11, dilution 1:1000), RIP1 (Rabbit IgG, clone D94C12, dilution 1:1000), pRIP1S166 (Rabbit IgG, clone D1L3S, dilution 1:1000), RIP3 (Rabbit IgG, clone E1Z1D, dilution 1:1000), pRIP3S227 (Rabbit IgG, clone D6W2T, dilution 1:1000), MLKL (Rabbit IgG, clone D2I6N, dilution 1:1000) and pMLKLS358 (Rabbit IgG, clone D6H3V, dilution 1:1000) and pRIP1 (Rabbit IgG, clone D8I3A, dilution 1:1000) (all Cell Signaling, Frankfurt am Main, Germany). HRP-conjugated rabbit-anti-mouse IgG or goat-anti-rabbit IgG served as secondary antibodies (IgG, both: dilution 1:1000, Dako, Glosturp, Denmark). …”
Supplementary Material section (lines 611-612):
“…Figure S6: Detailed information about Figure 10 - Protein expression profile of RIP1, pRIP1, RIP3, pRIP3, MLKL and pMLKL in parental and DX-resistant PC3 and DU145 cells,…”
